# Early Paleoindian use of canids, felids, and hares for bone needle production at the La Prele site, Wyoming, USA

Spencer R. Pelton[1]*, McKenna Litynski[2], Sarah A. Allaun[3], Michael Buckley[4], Jack Govaerts[5], Todd Schoborg[5], Matthew O'Brien[6], Matthew G. Hill[7], Paul Sanders[1], Madeline E. Mackie[8], Robert L. Kelly[2], Todd A. Surovell[2]

1 Office of the Wyoming State Archaeologist, University of Wyoming, Laramie, Wyoming, United States of America, 2 Department of Anthropology, University of Wyoming, Laramie, Wyoming, United States of America, 3 History Colorado Office of Archaeology and Historic Preservation, Denver, Colorado, United States of America, 4 School of Natural Sciences, Manchester Institute of Biotechnology, University of Manchester, Manchester, United Kingdom, 5 Department of Molecular Biology, University of Wyoming, Laramie, Wyoming, United States of America, 6 Department of Anthropology, California State University Chico, Chico, California, United States of America, 7 World Languages and Cultures, Iowa State University, Ames, Iowa, United States of America, 8 Department of Anthropology, Michigan State University, East Lansing, Michigan, United States of America

* spencer.pelton@wyo.gov

**Data Availability Statement:** All relevant data are within the paper and its Supporting Information files.

## Abstract

We report the first identifications of species and element used to produce Paleolithic bone needles. Archaeologists have used the tailored, fur-fringed garments of high latitude foragers as modern analogs for the clothes of Paleolithic foragers, arguing that the appearance of bone needles and fur bearer remains in archaeological sites c. 40,000 BP is indirect evidence for the advent of tailored garments at this time. These garments partially enabled modern human dispersal to northern latitudes and eventually enabled colonization of the Americas ca. 14,500 BP. Despite the importance of bone needles to explaining global modern human dispersal, archaeologists have never identified the materials used to produce them, thus limiting understanding of this important cultural innovation. We use Zooarchaeology by Mass Spectrometry (ZooMS) and Micro-CT scanning to establish that bone needles at the ca. 12,900 BP La Prele site (Wyoming, USA) were produced from the bones of canids, felids, and hares. We propose that these bones were used by the Early Paleoindian foragers at La Prele because they were scaled correctly for bone needle production and readily available within the campsite, having remained affixed to pelts sewn into complex garments. Combined with a review of comparable evidence from other North American Paleoindian sites, our results suggest that North American Early Paleoindians had direct access to fur-bearing predators, likely from trapping, and represent some of the most detailed evidence yet discovered for Paleoindian garments.

**Funding:** Funding for this project includes the National Science Foundation (award #1947297), the Wyoming Cultural Trust Fund, the Quest Archaeological Research Program at Southern Methodist University, the National Geographic Society, Ed and Shirley Cheramy, and the George C. Frison Institute for Archaeology and Anthropology. The funders had no role in study design, data collection and analysis, decision to publish, or preparation of the manuscript.

**Competing interests:** The authors have declared that no competing interests exist.

## Introduction

Thermoregulation was a fundamental constraint on the pacing and character of global modern human dispersal. Modern humans are primarily physiologically adapted to the climates of sub-Saharan Africa where much of their evolutionary lineage exists [1–4], so expansion into northern latitudes required the adoption of technologies that buffered early modern humans against increasingly severe cold conditions [5–9]. Of the various technologies and behaviors enacted to cope with cold temperatures, complex, tailored garments are among the most important.

Tailored garments are an improvement over draped garments because they adhere closely to the skin [10] and closely stitched seams provide a water and windtight barrier against the elements [11]. Once equipped with such garments, modern humans had the capacity to expand their range to places from which they were previously excluded due to the threat of hypothermia or death from exposure.

Little direct evidence exists for these sorts of garments (but see [12]), but indirect evidence for them includes bone needles and the bones of fur bearers. The rationale for assuming bone needles are an archaeological proxy for tailored garments is simple. Sewers produce tailored garments with intricate, tightly stitched seams, which require needles to assemble rather than awls, which create more widely-spaced and coarsely perforated seams [13]. Bone needles emerge in Eurasia beginning ca. 40,000 BP [8, 14–17] and in North American Paleoindian sites between ca. 12,000 and 13,000 BP [18, 19].

The bones of fur bearers have received less attention as an indirect proxy for tailored garment production, and the link between the phenomena is more complicated, having its basis in foraging theory. Fur bearers such as canids, felids, leporids, and mustelids possess pelts with soft, tightly spaced hairs that trap a layer of stationary air near the skin's surface [20]. These animals are typically ranked low in a diet breadth framework [21, 22] because they are small, difficult to hunt, and in the case of carnivores sparse on the landscape. Thus, their presence in archaeological assemblages is hard to explain from a purely dietary framework. In ethnographic settings, foragers often cull furbearers only for their pelts and only with the use of traps to limit search time [23, 24]. If their primary product is fur rather than meat, then furbearers might confound optimal foraging models as traditionally conceived based on kilocalorie value. Thus, archaeologists have explained their presence in Paleolithic contexts otherwise dominated by ungulate remains as a result of pelt acquisition [25–27].

Here, we present evidence for tailored garment production using fur bearers from the La Prele site (Fig 1). La Prele is an Early Paleoindian (ca. 12,900 cal BP) mammoth kill and campsite buried over 3 m deep on a tributary on the North Platte River near Douglas, WY associated with diagnostic artifacts of the Clovis and Folsom cultural complexes [28]. Ten seasons of excavations in four major blocks (A through D) have yielded tens of thousands of artifacts associated with a single occupation [29]. Among the wide diversity of artifacts recovered from the site thus far are 32 bone needle fragments. We use Zooarchaeology by Mass Spectrometry (ZooMS) and Micro-CT to argue that the Early Paleoindian foragers at La Prele had primary access to fur bearers, the bones from which they produced bone needles and the furs from which they incorporated into tailored garments.

## Materials and methods

The artifacts used in this study were recovered from archaeological excavations by the University of Wyoming Department of Anthropology between 2015 and 2022 from the La Prele site, located on private land in Converse County, Wyoming. The artifacts are housed at the

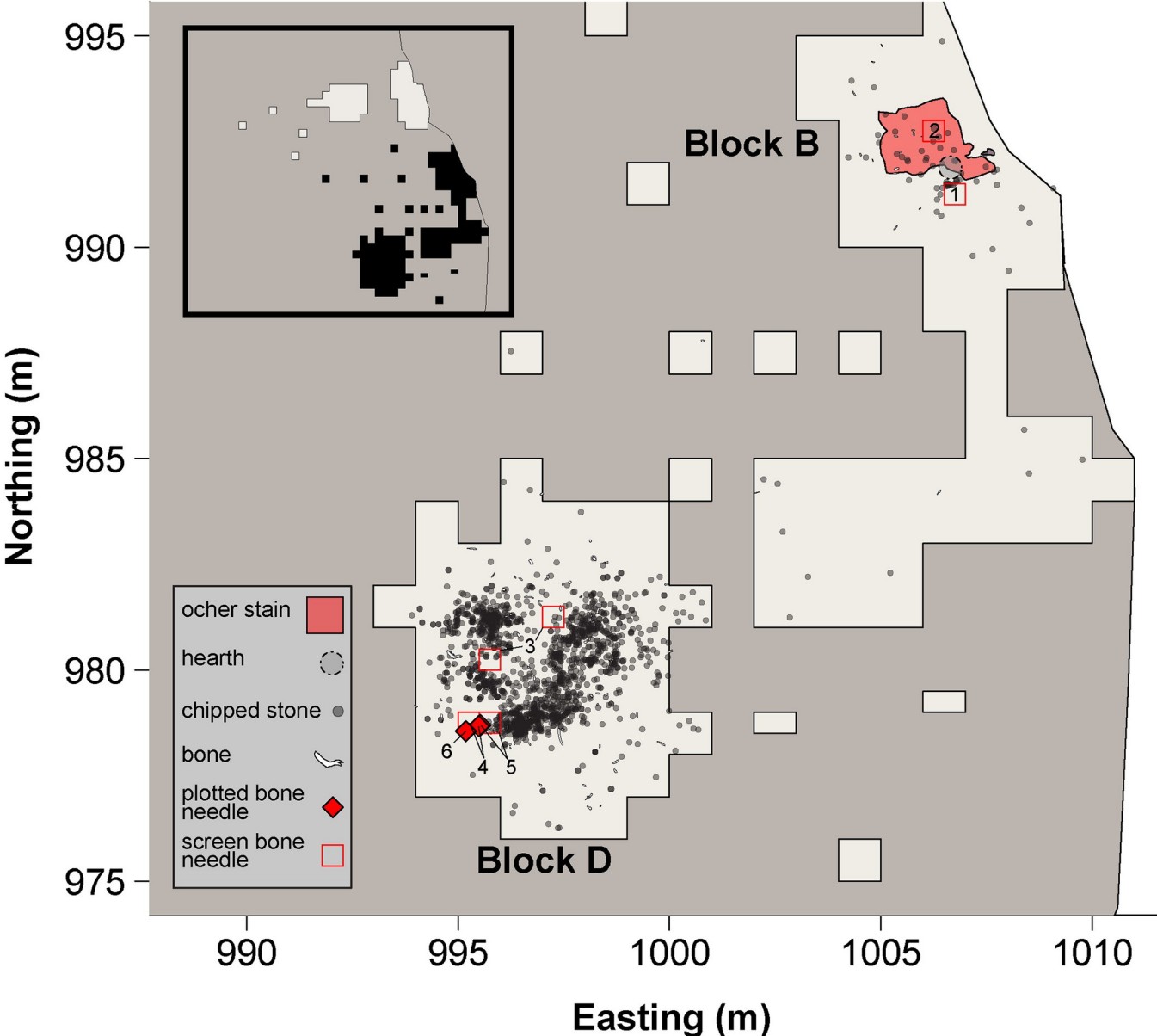

**Fig 1. Archaeological plan map of La Prele Blocks B and D showing the locations of recovered bone needle fragments.** Inset map shows entire site; black shaded polygons depict area of interest. Numbered bone needle provenience locations refer to bone needle ID's of Table 1.

University of Wyoming in the lab of Todd Surovell. No permits were required for the described study, which complied with all relevant regulations.

We obtained high resolution images and metric data for bone needles and comparative specimens using a ZEISS Xradia 610 Versa using the following X-ray tube parameters: 60 kV, 108 µA and 6.5 W of power. Images were collected using either the 4x objective (bone needle fragments) or the 0.4 × objective (animal bones), and a final pixel size of 4–10 µm. A total of 4501 projection images were acquired using 360 degrees of sample rotation. Scout and Scan™ Reconstructor (v16.1) was used to generate the final tomograms. We obtained metric attributes using Dragonfly software (Comet Technologies, v2022.2). For each comparative

specimen or bone needle fragment, we obtained two measurements (thickness and width) at five locations evenly spaced down the long axis of each specimen. We measured each specimen in five locations to characterize the range of variation present within each specimen, regardless of the specimen's total size.

For three archaeological samples (FS 2222, 4678, and 14770), we used minimally destructive methods for the extraction of soluble collagen [30, 31] at the University of Manchester's Ancient Biomolecules Laboratory. Collagen was solubilized in 0.3 M HCl and exchanged into 50 mM ammonium bicarbonate using a 10 kDa ultrafilter. After trypsin digestion, samples were analyzed with a Bruker Rapiflex MALDI TOF/TOF mass spectrometer. The remaining archaeological samples were processed using destructive methods [30, 32–34]. Small needle fragments (1 to 20 mg) were ground in a clean agate mortar and pestle. The bone powder was demineralized with 0.6 M HCl, and humic acids were extracted with 0.1 M NaOH. After gelatinization for 1 hour at 65°C in ammonium bicarbonate, samples were digested with porcine trypsin. Spectra were generated using a Sciex 5800 MALDI-TOF mass spectrometer in the Basile Lab at the University of Wyoming.

To identify taxa, we mostly relied on published marker peptides [32, 35, 36] in combination with MALDI-TOF spectra of tryptic digests run on species specifically included for this study. For the latter taxa, we recorded the presence or absence of known marker peptides in mass spectra (S1 File). Additional species analyzed for this study are *Miracinonyx trumani* (American cheetah-like cat), *Puma concolor* (mountain lion), *Smilodon fatalis* (saber-toothed cat), *Vulpes velox* (swift fox), *Uroycyon cinereoargenteus* (gray fox), *Aeonocyon dirus* (dire wolf), *Lepus californicus (*black-tailed jackrabbit), *Lepus americanus* (snowshoe hare), *Lepus townsendii* (white-tailed jackrabbit), *Sylvilagus audubonii* (desert cottontail), *Brachylagus idahoensis* (pygmy rabbit), and *Ochotona princeps* (American pika). Specimens of extant species were sampled from the modern zooarchaeology comparative collection at the University of Wyoming. We sampled a *Miracinonyx trumani* innominate *(*Cat no. 1750A*)* from Little Canyon Creek Cave (48WA323), Wyoming curated at the University of Wyoming Archaeological Repository. We sampled a *Smilodon fatalis* cranium (SUI 15–548) from Rancho La Brea and an *Aeonocyon dirus* radius (SUI-149737) from the West Noadaway River, Mills County Iowa. Both specimens are curated at the Paleontological Repository, University of Iowa, Iowa City. The *A. dirus* radius was collected by Jim McClarnon in May, 2019 in shallow water resting on the bed of the West Nodaway River, about 6 km north of the small town of Villisca, Montgomery County, Iowa, in the northeast quarter of Section 4, T71N-R36W, or 0.5 km north of the location where U.S. Highway 34 crosses the river. It was donated to the Paleontological Repository, Department of Earth and Environmental Sciences, University of Iowa (SUI), Iowa City, Iowa, on March 6, 2024 for permanent curation.

Using the nomenclature of Brown [37], the α2 793 marker peptide (D from Buckley et al. 2009) [32] was most reliable for identifying the family of origin of archaeological samples as shown in S1 File; peaks at 2129.1 m/z, 2131.1 m/z, and 2163.1 m/z were identified as Leporidae, Canidae, and Felidae, respectively. Additional identification details are provided below.

The presence of α2 978 ("peptide A") peaks at 1221.6/1235.6 m/z, a α2 793 peak at 2129.1 m/z (D), and a α1 586 peaks at 2883.4/2899.4 m/z (F) positively identified a sample as a lagomorph. The additional presence of an α2 454 peak at 2808.3 m/z (E) distinguishes hares from rabbits and pika, and that peptide occurs in all three archaeological specimens identified as derived from lagomorph bone. Therefore, the needle and preform fragments manufactured from lagomorph bone were derived from a jackrabbit or hare.

Carnivores were identified based on the presence of the α1 586 peptide pair at 2853.4/2869.4 m/z in combination with the α2 757 peaks at 2983.5/2999.5 m/z. Canids could generally be identified by the presence of the α2 978 peaks at 1210.7/1226.7 m/z, although those peptides

were not observed in gray fox (*Urocyon cinereoargenteus)* or dire wolf (*Aenocyon dirus*). Red fox (*Vulpes vulpes*) was identified by the presence of the α2 484 peptide at 1437.7 m/z. All needle specimens identified as having been manufactured from canid bone exhibit this peak and thus were positively identified as red fox.

For one sample (FS 4678), a felid identification could be made by presence of the α2 978 peptide at 1207.6/1223.6 m/z and the absence of the α2 793 peptide at 2131.1 m/z. Two additional felid identifications were made using the presence of the α2 793 marker. That peptide allowed discrimination between subfamily Felinae and subfamilies Pantherinae and Machairodontinae. The former show a peak at 2163.1 m/z and the latter at 2147.1 m/z. All feline needle specimens show the 2163.1 m/z marker peptide, and thus can be identified as having been derived from one of four possible taxa: bobcat (*Lynx rufus*), lynx (*Lynx canadensis*), mountain lion (*Puma concolor*), or American cheetah (*Miracinonyx trumani*). Mustelidae and Procyonidae can be eliminated as candidate taxa based on the absence of the 1219.6/1235.6 α2 978 peptide pair in all needle spectra.

## Results

We recovered 32 bone needle fragments from La Prele from 11 field specimen (FS) numbers representing 10 distinct provenience locations, including 17 from Block B and 15 from Block D (Fig 1; Table 1). Ten fragments from Block D are what we interpret as two bone needle preform fragments. The preform fragments are relatively wide (mean = 1.96 mm, SD = 0.48 mm, range = 1.06–2.91 mm) and square compared to finished bone needles but are visibly worked. Consistent with other Paleoindian bone needles [18], needle fragments are an average of 1.5 mm wide (mean = 1.45 mm; SD = .33 mm; range = 0.54–2.13 mm). All bone needle fragments are associated with dense accumulations of artifacts, likely the interiors of houses [38], and are typically discarded near concentrations of burned artifacts indicative of hearth-centered activity areas.

We attempted ZooMS on 11 bone needle fragments, 1 from each of the 11 bone needle FS numbers. We successfully obtained spectra from 10 specimens, including 4 from Block B and 6 from Block D. From these, we identified three families of mammalian fauna, including Canidae, Felidae, and Leporidae (Fig 2; S1 File). Block B contains needle fragments produced from canid and felid bone and Block D contains needle and preform fragments produced from canid and leporid bone.

**Table 1. Summary of bone needle artifacts from La Prele.**

| FS | ID | N | E | Z | block | type | width mean, SD, range (μ) | thickness mean, SD, range (μ) | count | ZooMS ID (genus) |
|----|----|------|------|------|-------|------|---------------------------|-------------------------------|-------|------------------|
| 2222* | 1 | 991.250 | 1006.750 | 97.300 | B | N | 1560 +/- 43; 1502–1627 | 1347 +/- 78; 1216–1459 | 3 | *Vulpes* |
| 4678* | 2 | 992.750 | 1006.250 | 97.350 | B | N | 1980 +/- 54; 1911–2057 | 1399 +/- 271; 872–1684 | 3 | *Lynx, Puma, or Miracinonyx* |
| 4957* | 2 | 992.750 | 1006.250 | 97.300 | B | N | 1751 +/- 103; 1563–1888 | 1666 +/- 117; 1468–1833 | 5 | *Lynx, Puma, or Miracinonyx* |
| 5854* | 2 | 992.750 | 1006.250 | 97.300 | B | N | 1222 +/-190; 876–1458 | 1051 +/-225; 539–1397 | 6 | *Lynx, Puma, or Miracinonyx* |
| 12378* | 3 | 980.250 | 995.750 | 97.400 | D | N | 2036 +/- 79; 1929–2129 | 1226 +/- 35; 1172–1265 | 1 | *Lepus* |
| 11543* | 3 | 981.250 | 997.250 | 97.500 | D | N | 1601 +/-41; 1538–1584 | 1312 +/- 31; 1283–1352 | 2 | *no result* |
| 14617 | 4 | 978.699 | 995.529 | 97.442 | D | P | 1632 +/-240; 1099–2164 | 882 +/- 309; 188–1394 | 6 | *Lepus* |
| 14770 | 5 | 978.673 | 995.489 | 97.431 | D | P | 1971 +/- 174; 1678–2157 | 1022 +/- 82; 819–1106 | 1 | *Vulpes* |
| 14726* | 5 | 978.750 | 995.750 | 97.400 | D | N | Small tip, not scanned | | 1 | *Vulpes* |
| 14477 | 6 | 978.554 | 995.182 | 97.451 | D | P | 2408 +/- 230; 2042–2911 | 1650 +/- 320; 1315–2407 | 1 | *Lepus* |
| 14511* | 4 | 978.750 | 995.250 | 97.450 | D | P | 2075 +/- 535; 1056–2701 | 1147 +/- 271; 592–1421 | 3 | Lepus |

*screen artifacts.

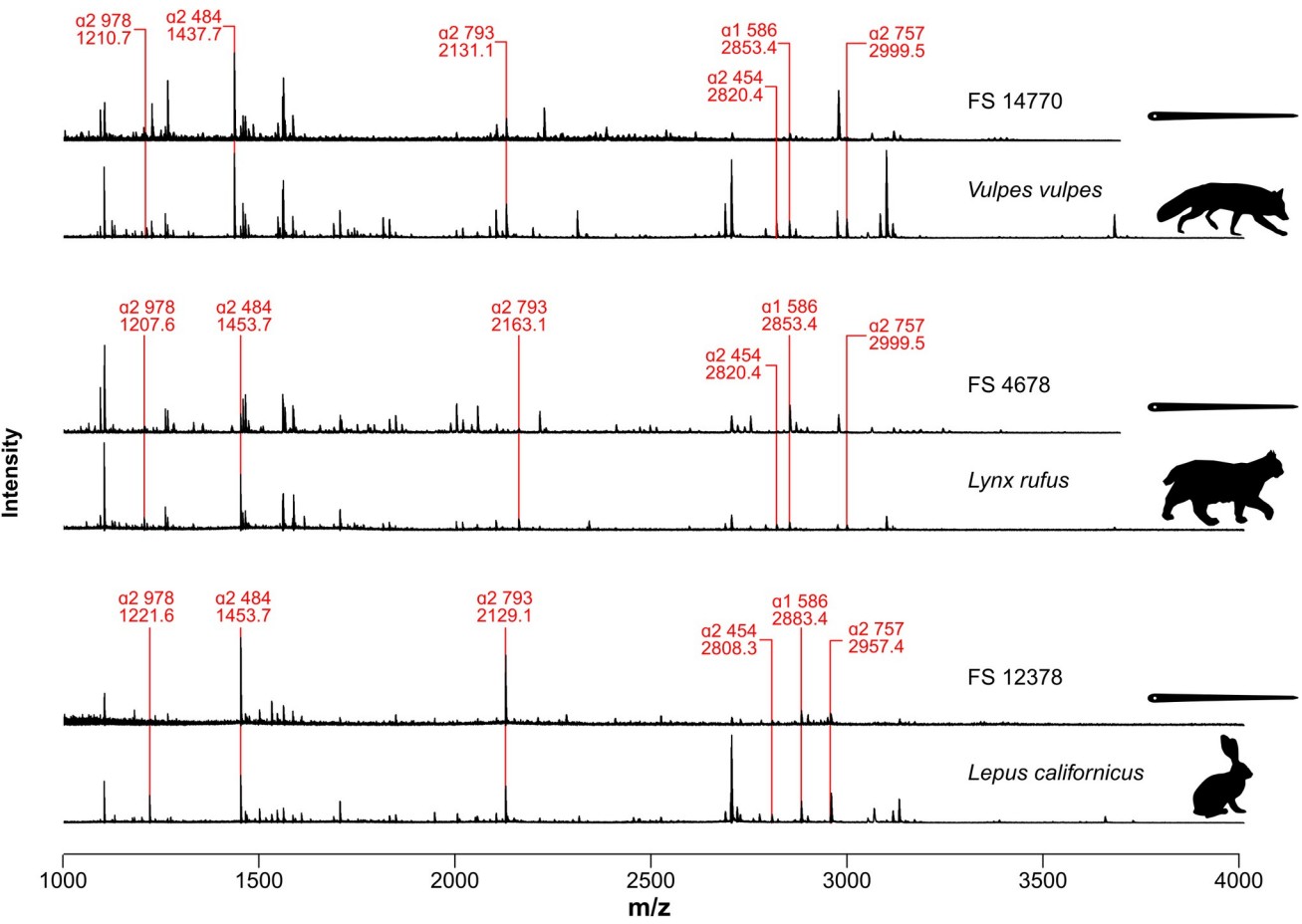

**Fig 2. MALDI-TOF spectra for three needle fragments in comparison to reference spectra for closely related taxa (red fox, bobcat, and black-tailed jackrabbit).** Needle FS 14770 (upper) is identified as *Vulpes vulpes*. Needle FS 4678 (middle) is identified as subfamily Felinae. Needle FS 12378 (lower) is identified as genus *Lepus*.

Possible canid species in Pleistocene Wyoming include the coyote (*Canis latrans*), gray wolf (*Canis lupus*), gray fox (*Urocyon cinereoargenteus)*, red fox (*Vulpes vulpes*), swift fox (*Vulpes velox*), the extinct Pleistocene Dire wolf (*Aenocyon dirus*), and possibly domesticated dogs (*Canis familiaris*) maintained by the La Prele site's inhabitants [39]. All canid bone needle fragments are positively identified as red fox (*Vulpes vulpes*) based on the presence of the α2 484 marker peptide peak at 1437.7 m/z [32].

Possible felid species in Pleistocene Wyoming include the Canada lynx (*Lynx canadensis*), bobcat (*Lynx rufus*), mountain lion (*Puma concolor*), saber-toothed cat (*Smilodon fatalis*), American cheetah-like cat (*Miracinonyx trumani*), Scimitar cat (*Homotherium* sp.), American lion (*Panthera atrox*), and Pleistocene North American jaguar (*Panthera onca augusta*). We have not compiled reference spectra for the Scimitar cat (*Homotherium*). Block B needle fragments are derived from the subfamily Felinae, meaning *Lynx* sp., *Puma concolor*, or *Miracinonyx trumani*.

Leporids in Wyoming include species of both hares (*Lepus* sp.) and rabbits (*Sylvilagus* sp.). Extant Wyoming hares are likely the same as those present during the Pleistocene and include the white-tailed jackrabbit (*Lepus townsendii*), the black-tailed jackrabbit (*Lepus californicus*), and the snowshoe hare (*Lepus americanus*). Rabbits include the desert cottontail (*Sylvilagus*

*audobonii*), mountain cottontail (*Sylvilagus nuttallii*), and likely the pygmy rabbit (*Brachylagus idahoensis*), though the latter's Pleistocene distribution in Wyoming is poorly understood. All four Block D leporid bone needles are hare (*Lepus* sp.), two of which we identified as bone needle preforms. All preform fragments are hare, suggesting that the La Prele site inhabitants of Block D were processing hare bones during needle production when the site was abandoned.

We estimate that La Prele has yielded a minimum number of four bone needles and two bone needle preforms based on a combination of their morphological characteristics, spatial distributions, and ZooMS results (Fig 3). We think Block B contains at least two bone needles. One Block B bone needle (ID-1) produced from *Vulpes* bone is represented by three fragments from a single 50 cm x 50 cm quadrant (quad) and 5 cm level. The other Block B bone needle (ID-2) produced from *Lynx/Puma/Miracinonyx* bone is represented by 14 fragments from a single quad dispersed across two levels.

We estimate that Block D contains at least two bone needle preforms and two finished bone needles. We discovered fragments of two preforms made of *Lepus* sp. bone (IDs 4 and 6) from the same quad and level, one complete preform plotted in situ (ID-6) and the other recovered in three pieces from the screen (ID-4). We suspect that one piece-plotted artifact represented by two conjoining fragments (FS 14617) may be a non-conjoining fragment of needle ID-4 because it could have been recovered from as close as 3 cm to it in the adjacent quad, appears similar in size and shape, and is also made of *Lepus* sp. bone.

We recovered 2 *Vulpes* needle fragments (ID-5) in the same location as the *Lepus* preforms (IDs 4 and 6). One fragment is a plotted bone needle eye (FS 14770) and the other is a bone needle tip recovered from the screen matrix of the adjacent quad (FS 14726). Given their context, these artifacts could have been as close as 1 cm apart, so we believe they are fragments of the same needle.

Finally, a needle midsection made of hare bone (FS 12378) and a needle fragment from near the distal tip that did not yield a ZooMS spectrum (FS 11543) are close enough in metric dimensions to be from the same artifact (ID-3), though they are located around 1.8 m apart. Thus, we conservatively believe they are part of the same artifact, acknowledging that they may be fragments of different artifacts.

Micro-CT scans of bone needle fragments indicate production from cortical bone. Although none of the bone needles from La Prele are complete due to use-related [18] and post-depositional fracturing, rare examples of mostly complete bone needles of comparable age from the Agate Basin [40] and Lindenmeier [41] sites are 5 to 7 cm long and 1.5 to 2.5 mm thick. The metapodials and long bones of canids, felids, and leporids are among the only bones present in their respective skeletons that meet these criteria (Fig 4), others being too curved (e.g., ribs), thin (e.g., scapulae), short (e.g., phalanges), or inappropriately shaped (e.g., vertebrae, innominate, or crania). Metapodial cortical bone thickness is especially similar to bone needle thickness. One would need only to score metapodials lengthwise with a flake or graving tool, split them, and then pare down their width through carving or abrading to achieve appropriate dimensions for bone needles. Thus, we think the most likely skeletal elements used to produce the La Prele bone needles were the metapodials of canids and felids and specifically the longer metatarsals of hares.

## Discussion

Bone needles are a well-established archaeological phenomenon associated with the North American Younger Dryas chronozone (ca. 12,900–11,600 cal BP) [18, 19], when foragers are assumed to have produced tailored garments in response to the return of a near glacial climate. For instance, as few as four Folsom complex (ca. 12,900 to 12,350 cal BP) [42] campsites have

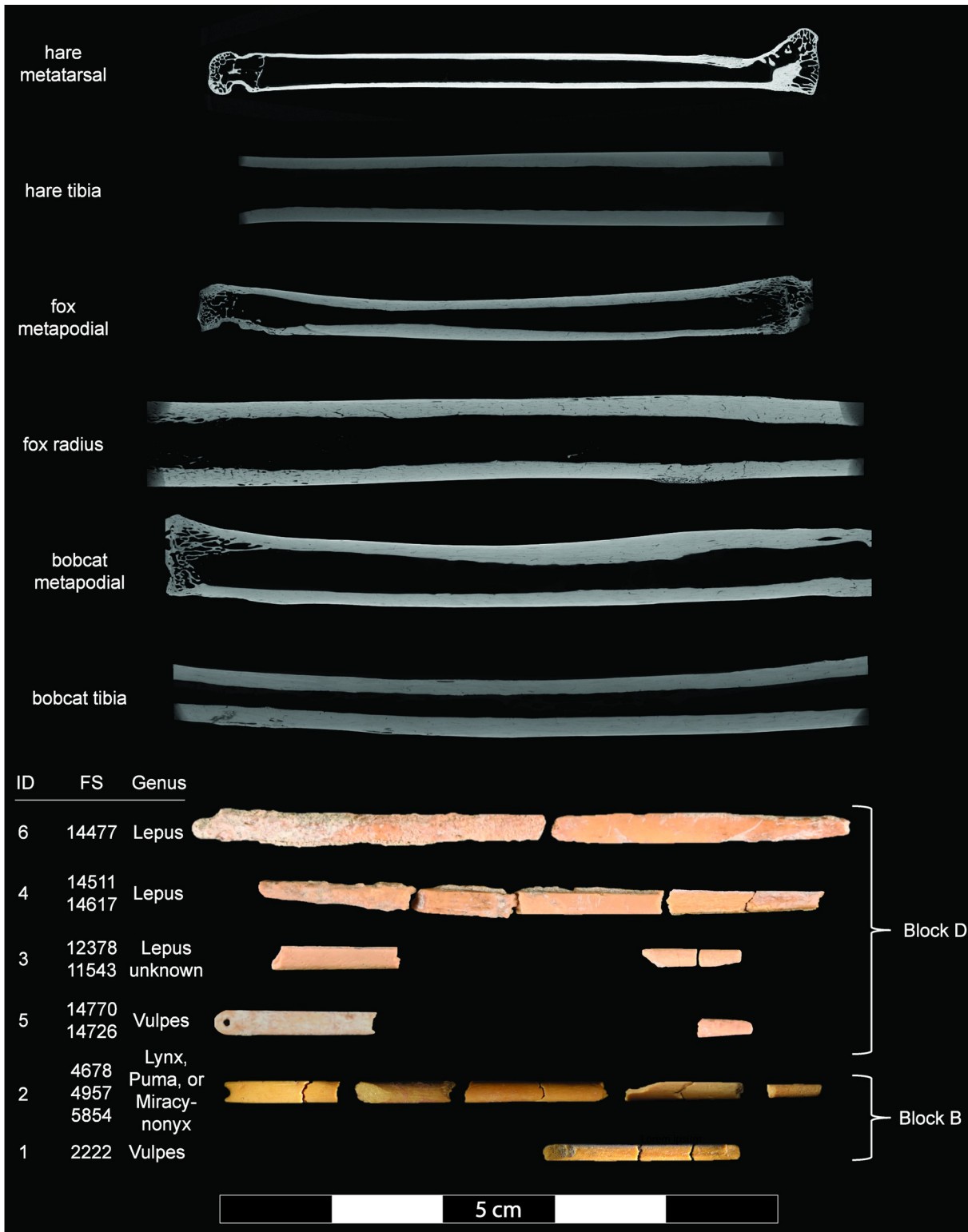

**Fig 3. Bone needle and needle preform reconstructions and Micro-CT scans of comparative faunal specimens.**

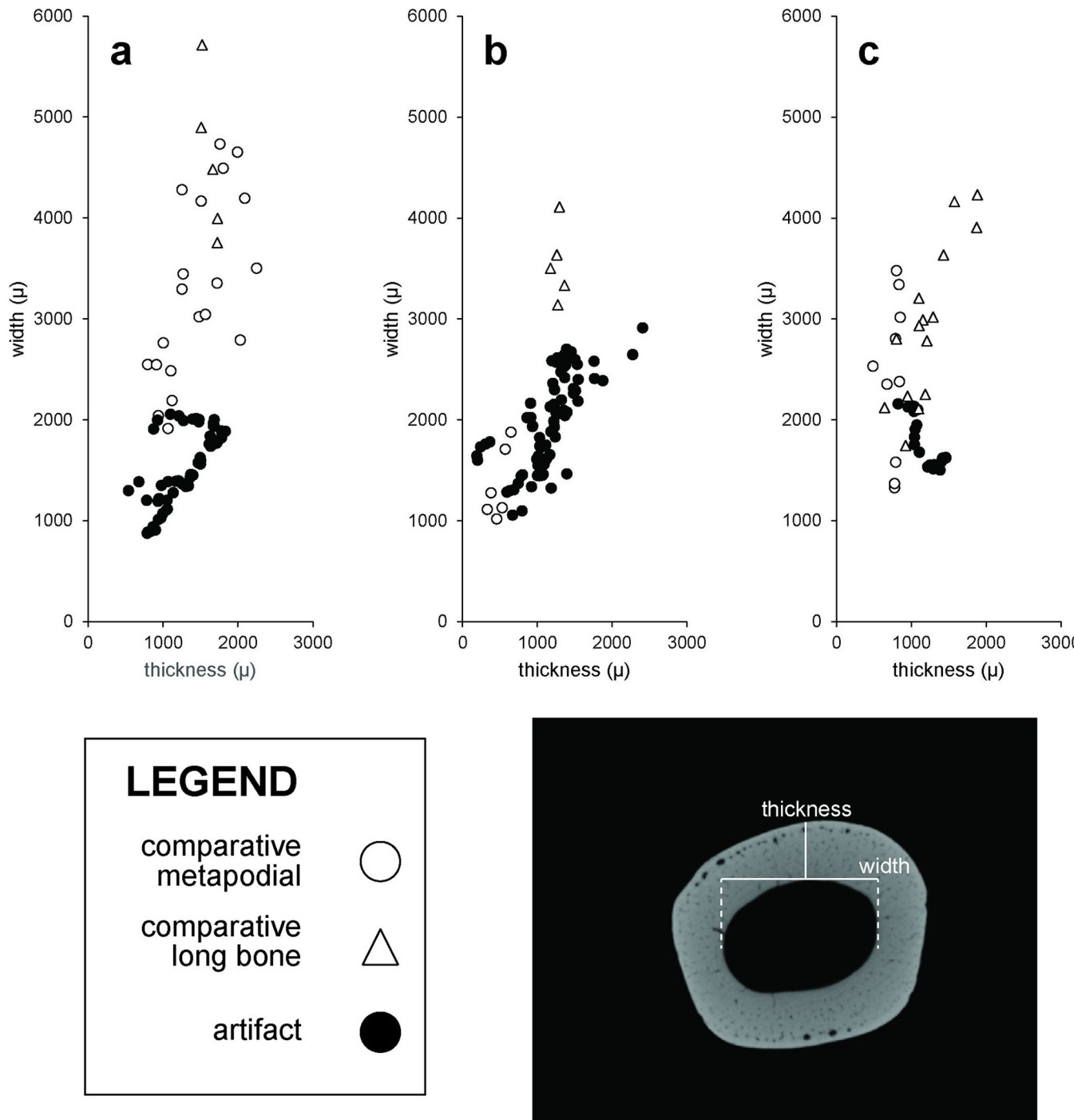

**Fig 4. Metric comparisons between bone needle artifacts and comparative faunal specimens for La Prele.** a) felid needles and *Lynx rufus*, b) *Lepus* needles/ needle preforms and *Lepus americanus*, and c) *Vulpes* needles and *Vulpes vulpes*.

been excavated in a way likely to yield bone needles (i.e., water screened through 1/16th inch mesh), yet at least five have produced them (including La Prele) [6]. Archaeologists have found bone needles in the campsites and burials of other Younger Dryas-aged cultures throughout North America as well [19]. Given the right preservation conditions and recovery

techniques, bone needles are seemingly ubiquitous in Younger Dryas-aged campsites in North America.

Additionally, a relatively large number of Early Paleoindian (ca. 13,200–12,000 cal BP) sites contain the bones of fur-bearing carnivores and lagomorphs. In a compilation of 36 Clovis (ca. 13,200–12,700 cal BP) faunal assemblages, six (17%) contain canid remains, one of which also contains felid (*Smilodon*), though the *Smilodon* is likely not associated with human activity [43]. In a compilation of 17 Folsom campsites that contain preserved bone, six (35%) contain the bones of fur-bearing carnivores, including gray wolf (*Canis lupus*), coyote (*Canis latrans*), swift fox (*Vulpes velox*), red fox (*Vulpes vulpes*), domestic dog (*Canis familiaris*), and bobcat (*Lynx rufus*) [6]. Lagomorphs are even more common in Early Paleoindian sites, though their association with human activity is not always certain due to their natural abundance on the landscape and tendency to burrow into archaeological deposits. Around 31% of Clovis [43] and 73% of Folsom [44] faunal assemblages contain lagomorph bones, including La Prele [45], and they are overwhelmingly *Lepus* sp. when identifiable. In fact, lagomorphs are the second most common animal in Folsom sites next to bison. Evidently, Early Paleoindians often obtained fur-bearing animals, and this tendency appears more common in Younger Dryas-aged sites (i.e., Folsom-aged) when the use of pelts in tailored garments would have been more common.

Results from La Prele are consistent with this larger pattern of bone needles and fur-bearing mammals being associated with the Early Paleoindian period. At La Prele, the production of bone needles from fur-bearing mammal bone most likely occurred because both bone needles and fur-bearer pelts were used to produce tailored garments. We suspect that fur-bearer foot bones were available to sewers at La Prele due to the way in which fur-bearer pelts are commonly removed from animals, which typically leaves foot bones attached [26, 27, 40]. In need of a needle, sewers at La Prele could have simply removed a metapodial from a nearby pelt, either sewn into a garment or awaiting to be, split it, and then abraded the bone to a width of 1 to 3 mm for use.

By extension, these results are compelling evidence that the earliest North Americans routinely trapped game. Most fur-bearing carnivores possess small return rates during encounter-based hunting due to their small size, low population densities, and wary avoidance behaviors [24, 46], as do lagomorphs outside of large communal drives [47] for which no evidence exists in North America prior to the Early Holocene [48, 49]. However, those poor returns are mitigated through trapping, which diminishes the search time required to pursue these types of game. In a recent review of all well-established return rates in the ethnographic literature, Morin and colleagues [24] document relatively few fur bearers, and those present were invariably trapped or hunted in game drives rather than pursued in a solo hunting scenario. Given that ethnographic cases of solo, encounter-based hunting for fur bearers appear to be rare, we must assume that such animals were rarely if ever obtained in ways other than trapping or driving (in the case of lagomorphs). Moreover, spears with large points seem like impractical weapons for hunting small carnivores and hares. While game trapping is widely accepted for the Eurasian Paleolithic [50], it is not typically acknowledged for the North American Paleoindian record. Given that trapping technologies were maintained through North American colonization, we find it most likely that Early Paleoindians operated traplines, at least seasonally, to obtain fur-bearing animals alongside the hunts used to procure large game whose bones dominate Early Paleoindian archaeological sites [43]. A further expectation of trapping is an over-representation of males among fur-bearing carnivores in archaeological sites, which are more likely to be trapped given behavioral differences [51]. Sex determination of fur bearing carnivores in Paleoindian assemblages would thus be an interesting avenue for future research.

## Conclusions

Our study is the first to identify the species and likely elements from which Paleoindians produced eyed bone needles. Bone needle production techniques have until this point been largely theoretical, based on incised bone objects argued to represent bone needle preforms [52, 53] and experimental needle production [54, 55]. We were able to clarify some aspects of bone needle production techniques through the combined use of ZooMS and Micro-CT scanning, the first applications of both of these technologies to the study of Paleoindian bone needles (but not beads [56]).

Our results are strong evidence for tailored garment production using bone needles and fur-bearing animal pelts. Such garments might have looked comparable to those of the Inuit, who sewed fur bearer pelts into the fringes of parkas whose base material was typically comprised of ungulate hide [11] and used them for hats and mittens. The cold conditions of the North American Younger Dryas in northerly latitudes likely inspired a greater reliance on such garments, and the sparse Early Paleoindian archaeological record suggests a relative abundance of bone needles and fur bearers in Younger Dryas-aged sites relative to periods before and after.

Finally, our results have some relevance to the debate surrounding Early Paleoindian subsistence. The presence of fur bearers in Early Paleoindian sites has been argued to support the existence of a 'broad spectrum' subsistence base characterized by generalist rather than specialist diets [57, 58]. Our results are a good reminder that foragers use animal products for a wide range of purposes other than subsistence, and that the mere presence of animal bones in an archaeological site need not be indicative of diet.

## Supporting information

**S1 File. MALDI-TOF spectra used to determine artifact taxa present in three Figures and a Table.**
(PDF)

## Acknowledgments

The Strock family has been gracious in allowing investigations of La Prele on their land. We thank the Science Initiative and the Center for Advanced Scientific Instrumentation at the University of Wyoming for use of the Zeiss Xradia Versa 610

## Author Contributions

**Conceptualization:** Spencer R. Pelton, Matthew O'Brien, Madeline E. Mackie, Robert L. Kelly, Todd A. Surovell.

**Formal analysis:** Spencer R. Pelton, McKenna Litynski, Sarah A. Allaun, Michael Buckley, Jack Govaerts, Todd Schoborg, Matthew O'Brien, Paul Sanders, Todd A. Surovell.

**Methodology:** Spencer R. Pelton.

**Project administration:** Spencer R. Pelton, Todd A. Surovell.

**Resources:** Matthew G. Hill.

**Visualization:** Spencer R. Pelton, Todd A. Surovell.

**Writing – original draft:** Spencer R. Pelton.

**Writing – review & editing:** Spencer R. Pelton.

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
