## [Decision Letter · Decision Letter 0]

18 Sep 2024

PONE-D-24-35763Early Paleoindian use of canids, felids, and hares for bone needle production at the La Prele site, Wyoming, USAPLOS ONE

Dear Dr. Pelton,

Thank you for submitting your manuscript to PLOS ONE. After careful consideration, we feel that it has merit but does not fully meet PLOS ONE’s publication criteria as it currently stands. Therefore, we invite you to submit a revised version of the manuscript that addresses the points raised during the review process.

This is an excellent study that will be of interest to many researchers. The two reviews of this paper are both very positive and raise only minor questions and suggest small edits. The two reviews conflict on the necessity of the description in the Material and Methods section, with Reviewer 1 suggesting it could be moved to Supplemental materials and Reviewer 2 requesting the addition of some detail in this section. My inclination is to keep the Materials and Methods section as is and perhaps address Reviewer 2's concern. 

We look forward to receiving your revised manuscript.

Kind regards,

Briggs Buchanan, Ph.D.

Academic Editor

PLOS ONE

2. In your manuscript, please provide additional information regarding the specimens used in your study. Ensure that you have reported human remain specimen numbers and complete repository information, including museum name and geographic location.

For more information on PLOS ONE's requirements for paleontology and archeology research, see https://journals.plos.org/plosone/s/submission-guidelines#loc-paleontology-and-archaeology-research.

“Funding for this project includes the National Science Foundation, the Wyoming Cultural Trust Fund, the Quest Archaeological Research Program at Southern Methodist University, and the George C. Frison Institute for Archaeology and Anthropology.”

“The Strock family has been gracious in allowing investigations of La Prele on their land. Funding for this project includes the National Science Foundation, the Wyoming Cultural Trust Fund, the Quest Archaeological Research Program at Southern Methodist University, and the George C. Frison Institute for Archaeology and Anthropology.”

“Funding for this project includes the National Science Foundation, the Wyoming Cultural Trust Fund, the Quest Archaeological Research Program at Southern Methodist University, and the George C. Frison Institute for Archaeology and Anthropology.”

Additional Editor Comments:

This is an excellent study that will be of interest to many researchers. The two reviews of this paper are both very positive and raise only minor questions and suggest small edits. The two reviews conflict on the necessity of the description in the Material and Methods section, with Reviewer 1 suggesting it could be moved to Supplemental materials and Reviewer 2 requesting the addition of some detail in this section. My inclination is to keep the Materials and Methods section as is and perhaps address Reviewer 2's concern.

Reviewers' comments:

Reviewer's Responses to Questions

**Comments to the Author**

1. Is the manuscript technically sound, and do the data support the conclusions?

Reviewer #1: Yes

Reviewer #2: Yes

2. Has the statistical analysis been performed appropriately and rigorously? 

Reviewer #1: N/A

Reviewer #2: N/A

3. Have the authors made all data underlying the findings in their manuscript fully available?

Reviewer #1: Yes

Reviewer #2: Yes

4. Is the manuscript presented in an intelligible fashion and written in standard English?

Reviewer #1: Yes

Reviewer #2: Yes

5. Review Comments to the Author

Reviewer #1: What a nifty paper!! Really solid analyses and logic that not only provide insight to which animal taxa were exploited for what purposes but reminds us quite explicitly (and correctly) that animals were more than food sources for foragers. I like this paper a lot and find it quite worthy of publication in PLOS One. However, before publication there are some things that need to be fixed; these are mostly typographical errors and points of clarification, nothing terribly onerous. Taking them in order in the manuscript:

--p. 3, last paragraph: “like canids, felids, etc”. which implies not canids, felids, etc. but something “like” them, so change “like” to “such as”

--p. 3, last paragraph, sentence spanning lines 4-6: may want to add that carnivores are relatively rare on the landscape whereas leporids are typically relatively abundant

--p. 3, last paragraph, third line from bottom: “the first evidence” when I initially read this, my thought was hardly that this was “the first” as others had made similar arguments, but it is of course the first discussion of La Prele needles (so far as I know). But think about deleting “the first” or rewording so the wrong impression is not suggested.

--p. 4, first paragraph under “Materials and Methods”, last sentence: why measured in five locations? Even on specimens a few mm long? And how are these data reflected in Fig. 4? Of what significance is taking measurements in five locations on a specimen, even if it is a complete needle? Without answering these questions, this is information that adds nothing but words to the manuscript.

--pp. 4-6: “Materials and Method”— I find much of these paragraphs to be unnecessary. ZooMS is a well established, well known and generally accepted viable method, and Micro-CT scanning seems similarly known among at least some people. These paragraphs distract one’s attention from the focus of the article—which taxa contributed their bones to Paleoindian needles? I suggest considering putting much of this in supplemental material, or shortening it considerably.

--p. 5, second paragraph: these taxonomic names should include the common names here (on first mention) as well. Will most readers know what a Miracinonyx trumani is? Or what a Uroycyon cinereoargenteus is? Once common names have been listed, any later mention of specific taxa can just be to taxonomic names or common names (one or the other). Listing both taxonomic and common names later (such as found on p. 6 and p. 8 and p. 11) is too late to be helpful (and duplication is unnecessary). See also my comment on p. 7 below.

--p. 6, next to last line: “std” I suspect I know what this is--it likely is not “sexually transmitted disease”. But from my perspective it is not standard statistical abbreviation; SD is what I am used to seeing (assuming what is meant is “standard deviation”). Same comment for second line on p. 7.

--p. 7, table 1: why are both species and common names listed? I suggest using one or the other rather than the mixture. See my comment on p. 5 above.

--p. 8, last paragraph, third line: “sp.” should not be italicized here or anyplace else.

--p. 8, last line: “poorly understood”. This may be true in Wyoming but not everywhere where pygmy rabbit is found (e.g., Lyman, RL. 2004. Biogeographic and Paleoenvironmental Implications of Late Quaternary Pygmy Rabbits (Brachylagus idahoensis) in Eastern Washington. Western North American Naturalist 64:1–6). Does perhaps the most recent rendition of FAUNMAP have some info on this?

--p. 9, line 4: “four bone needles”. I suspect, based on subsequent discussion, this is much like a “minimum number” estimate based on similarity in dimensions, provenience, and taxon represented by sets of fragments. It would help to state such bases for the estimate here rather than leaving it until later.

--p. 9, just above figure 3 caption: “Lynx/Puma/Miracinonyx” all should be italicized

--p. 10, second paragraph, line 3: “5 to 7 cm long”—how many specimens are complete or nearly so? As I recall from Lyman (2015), most specimens are shorter but are incomplete. And think about this (as Lyman 2015 suggests), if the needle is two long, it likely will snap as one attempts to wiggle it back and forth through a too-small hole already in the hide. Why do you suppose so many of the needles we find are broken? I doubt this is all post-depositional fragmentation.

--p. 10, second paragraph, last four lines: lengthwise split leporid metapodials. Not an unreasonable suggestion, but have you tried to split one of them lengthwise? I suspect it would not be easy to do it consistently successfully. How, for instance, is the metapodial firmly secured such that graving its length (seems the most logical technique of splitting) doesn’t skitter all over?

--p. 11, fourth line from bottom: this took me a few minutes to figure out what the “two artifacts” are: the bone needle artifact was easy to note, but the “fur-bearing mammal bone” as an artifact was not, likely because 99.3% of all zooarchaeological mammal bones I have ever studied were NOT considered artifacts by the archaeologists who dug them up and sent them to me. And this is my impression of the discipline at large—mammal bones ain’t artifacts (if anything other than animal bones, they are ecofacts). Consider rewording this sentence.

--p. 12, lines 4-5: you might consider noting that the trapping notion has a test implication. Trapped carnivorous fur bearers tend to be dominated by males rather than females given behavioral differences (for references and an archaeological example, see: Lyman, R. L. 2007. Prehistoric Mink (Mustela vison) Trapping on the Northwest Coast (USA). Journal of Field Archaeology 32:91–95). Not that you need to do this, but it might be good to mention this as an avenue of future research.

--p. 12, line 3 of conclusions: “entirely conjectural” seems an overstatement as it implicitly derogates and belittles experimental work. Experimental work of course does not show how things were done in the past, only how they may have been done efficiently and effectively. But to categorize that work as conjectural will no doubt offend lithic replicators and any other experimental archaeologist. To then turn around and say “we obtained these results” is not only confusing (which results? On manufacturing protocol?) but misleading as it implies you did something regarding “bone needle production techniques” that previous workers did not do. In fact, you only determined which taxa were exploited; you did not determine how the needles were manufactured. And, you only conjectured that leporid metapodials were split lengthwise, and you did not even do experimental work to show that such was possible or whether grinding/abrasion or whittling or whatever was used to finish the needle. And you mention no evidence on the archaeological specimens as to whether they were ground/abraded or whittled or what.

--p. 13-17: there are typographical errors (e.g., refs 1 &19), formatting inconsistencies (e.g., sometimes book titles are capitalized, sometimes not), and missing info (e.g., refs. 28 & 30 & 34) scattered among the references. I have not listed them all here.

The list of things I believe require attention may seem long, but as indicated above, they are all relatively minor things. They do, nevertheless, need to be addressed. Once they are, this will be a significant contribution and I look forward to seeing a revised version published. I wish to be named as referee: R. Lee Lyman

Reviewer #2: This is an excellent manuscript and addresses an area worthy of investigation. What's more, the data and arguments are presented clearly and concisely.

The paper would be improved by addressing/clarifying two minor issues. First, there are frequent references to prey choice models and the place of furbearers in those models. In the text it isn't clear if the traditional return rates on furbearers (and thus their rank in traditional diet breadth models) are based on hunting or trapping. I don't think this changes the outcome of the study in any way but it would clarify whether or not the trapping behavior proposed in the article would affect the rank of these resources in existing prey choice models.

Second, the methods indicate that taxonomic identification relied on published marker peptides plus additional taxa analyzed for this study. What isn't clear is whether the published marker peptides included spectra for mustelids, procyonids, and other small mammals that could potentially be represented, or whether these other taxa were eliminated by other means.

Perhaps these issues go without saying to someone more familiar with Zooms, but the average reader (like this reviewer) might benefit from clarification. Both of these issues could be easily clarified in an added sentence or two.

Well done, very interesting, and a worthwhile contribution.

6. PLOS authors have the option to publish the peer review history of their article (what does this mean?). If published, this will include your full peer review and any attached files.

Reviewer #1: **Yes: **R. Lee Lyman

Reviewer #2: **Yes: **David Kilby

---

## [Editor Report · Decision Letter 1]

29 Oct 2024

Early Paleoindian use of canids, felids, and hares for bone needle production at the La Prele site, Wyoming, USA

PONE-D-24-35763R1

Dear Dr. Pelton,

We’re pleased to inform you that your manuscript has been judged scientifically suitable for publication and will be formally accepted for publication once it meets all outstanding technical requirements.

Kind regards,

Briggs Buchanan, Ph.D.

Academic Editor

PLOS ONE
---

## [Editor Report · Acceptance letter]

1 Nov 2024

PONE-D-24-35763R1 

PLOS ONE

Dear Dr. Pelton, 

I'm pleased to inform you that your manuscript has been deemed suitable for publication in PLOS ONE. Congratulations! Your manuscript is now being handed over to our production team.

Kind regards, 

on behalf of

Dr. Briggs Buchanan 

Academic Editor

PLOS ONE